# Epigenetic Modulation with 5-Aza-CdR Prevents Metabolic-Associated Fatty Liver Disease Promoted by Maternal Overnutrition

**DOI:** 10.3390/nu17010106

**Published:** 2024-12-30

**Authors:** Henghui Cheng, Jie Wu, Hui Peng, Jiangyuan Li, Zhimin Liu, Xian Wang, Ke Zhang, Linglin Xie

**Affiliations:** 1Department of Nutrition, Texas A&M University, College Station, TX 77843, USA; hhcheng2007@hust.edu.cn (H.C.); huipeng@tjh.tjmu.edu.cn (H.P.); liuzhm3@mail.sysu.edu.cn (Z.L.); wangxian18@tamu.edu (X.W.); 2Center for Epigenetics & Disease Prevention, Institute of Biosciences & Technology, College of Medicine, Texas A&M University, Houston, TX 77030, USA; jiewu2014@whu.edu.cn (J.W.); jiangyuanli7@gmail.com (J.L.); 3Department of Statistics, Texas A&M University, College Station, TX 77843, USA

**Keywords:** obesity, MAFLD, DNA methylation, 5-Aza-2′-deoxycytidine (AZA)

## Abstract

Background/Objectives: This study builds on previous findings from mouse models, which showed that maternal overnutrition induced by a high-fat diet (HFD) promotes metabolic-associated fatty liver disease (MAFLD) in offspring, linked to global DNA hypermethylation. We explored whether epigenetic modulation with 5-Aza-CdR, a DNA methylation inhibitor, could prevent MAFLD in offspring exposed to maternal overnutrition. Methods: The offspring mice from dams of maternal overnutrition were fed either a chow diet or a high-fat diet (HFD) for 10 weeks. These mice were randomly divided into two groups: HFD, and AZA + HFD. Mice assigned to the AZA group were given 5-Aza-CdR during the last three weeks. Results: Our findings show that 5-Aza-CdR treatment in HFD-fed offspring effectively countered weight gain, improved glucose regulation, and minimized hepatic fat buildup along with serum lipid imbalances. Additionally, it boosted AMPK signaling and raised PPAR-α expression, pointing to enhanced fatty acid oxidation. We also detected an increase in JNK signaling, affecting the gene expression associated with cell death and proliferation. Notably, treated mice displayed more hepatic inflammation than the HFD group alone, suggesting a complex, dual impact on MAFLD management. Significant apoptotic and inflammatory gene changes were identified, along with corresponding differentially methylated regions triggered by 5-Aza-CdR, marking potential therapeutic targets. Conclusions: 5-Aza-CdR was shown to mitigate MAFLD features in offspring of maternal overnutrition by reversing DNA hypermethylation and improving metabolic pathways, though its dual impact on inflammation highlights the need for further research to optimize its therapeutic potential.

## 1. Introduction

Up to now, the significant increase in global obesity rates and related health issues has had a substantial adverse effect on human well-being and healthcare resources [1]. Metabolic-associated fatty liver disease (MAFLD), often described as the liver manifestation of metabolic syndrome, is associated with obesity, high insulin levels, and insulin resistance. It affects an estimated 10–24% of the general population across various regions globally. Notably, the prevalence of MAFLD can be as high as 75% among individuals with obesity [2,3]. These facts underscore the pressing requirement for effective preventive and treatment strategies to address obesity, which is a significant public health concern.

Research shows that both undernutrition and overnutrition during pregnancy and early childhood can significantly alter growth patterns, shifting offspring towards a higher risk of disease later in life [4,5,6,7,8,9,10,11]. Children born to mothers with pre-pregnancy obesity are at an increased risk of developing obesity themselves [12]. The concept of the “first 1000 days”—spanning from conception through the first two years of life—has emerged as a critical period for obesity prevention efforts [13]. The maternal effects on offspring obesity are likely attributable to epigenetic mechanisms including changing DNA methylation levels of important CpG loci associated with the expression of genes involved in energy metabolism such as PPARα, Cpt1α, and Lepr in the offspring [14,15,16,17,18,19,20,21,22,23,24,25,26,27,28,29,30].

Research, including our own, has consistently reported global DNA hypermethylation in offspring with MAFLD linked to maternal high-fat diets (HFD) [14,17,31]. In our study, we observed that reversing this hypermethylation by switching the maternal diet from an HFD to a normal-fat diet before pregnancy effectively prevented MAFLD in the offspring [14]. This finding suggests that maternal HFD-induced MAFLD may result from DNA methylation changes affecting multiple genes rather than one gene. As a cytosine analog, 5-Aza-CdR is known for its ability to inhibit DNA methyltransferase (DNMT) activity by forming a covalent complex with DNMTs and 5-Aza-CdR-substituted DNA, thereby reducing DNA methylation levels [32,33]. To explore this further, we investigated the impact of modifying DNA methylation on a genome-wide scale using the DNMT inhibitor 5-aza-2ʹ-deoxycytidine (5-Aza-CdR).

## 2. Materials and Methods

### 2.1. Animal Study

Before pregnancy, female mice with a mixed B6/129/SvEv genetic background were randomly placed on a high-fat diet (Research Diets, D12492, with 60% calories from fat) (Research Diets Inc., New Brunswick, NJ, USA) for nine weeks, continuing through pregnancy and lactation. After weaning, male offspring were split into two groups randomly: the HFD group received a high-fat diet for 10 weeks (*n* = 10 male offspring from 5 litters), while the HFD + AZA group followed the same diet but received 5-Aza-CdR injections (50 µg/mL, 0.25 mg/kg) from weeks 11 to 13 (*n* = 10 male offspring from 5 litters). All procedures were conducted following a protocol approved by the University of North Dakota’s Institutional Animal Care and Use Committee (The AUP No. IACUC 2016-0326), in compliance with the U.S. Public Health Service policies on humane animal research.

### 2.2. Antibodies

The PPAR-α antibody was sourced from Millipore Sigma (St. Louis, MO, USA), while antibodies targeting AKT, phosphorylated AKT, AMPK-β, phosphorylated AMPK-β, JNK, phosphorylated JNK, NF-κB, phosphorylated NF-κB, and GAPDH were obtained from Cell Signaling Technology (Danvers, MA, USA).

### 2.3. Tissue Histology and Immunostaining

Organ tissues were fixed in 4% paraformaldehyde at 4 °C overnight, dehydrated with FLEX 100 series, and embedded in paraffin. Sections (5 µm) were cut with a Leica RM2235 microtome and stained with H&E for evaluation using the Kleiner scoring system. FFPE liver samples were deparaffinized, rehydrated, treated for antigen retrieval, and incubated with the primary F4/80 antibody (1:200) for 2 h, followed by a secondary antibody (Alexa 488, 1:800) for 30 min, with DAPI used for nuclear counterstaining.

### 2.4. Intraperitoneal Injected Glucose Tolerance Test (IPGTT)

Mice underwent a 12 h overnight fast before an intraperitoneal glucose tolerance test (IPGTT) the following morning. For the test, 20% D-glucose in 0.9% saline was administered at a dose of 2.0 g/kg body weight.

### 2.5. Hepatic Triglyceride (TG) Measurement

Frozen liver tissues were homogenized in the lysis buffer, following the manufacturer’s instructions for the Triglyceride Colorimetric Assay Kit (Cayman Chemical, Ann Arbor, MI, USA).

### 2.6. Serum Cholesterol Measurement

Serum samples were collected, stored at −80 °C, and later analyzed for total cholesterol, low-density lipoprotein (LDL)/very low-density lipoprotein (VLDL), and high-density lipoprotein (HDL) using an assay kit from Abcam (Cambridge, UK). The assay demonstrated high accuracy with a coefficient of variation (CV) of less than 5%, ensuring reliable and reproducible results for lipid profiling, according to the manufacturer’s instructions.

### 2.7. ELISA for IRS2 and NPR-C

Commercially available kits for hepatic IRS2 (Aviva Systems Biology, San Diego, CA, USA) and serum NPR-C (ABclonal, Woburn, MA, USA) were used according to the manufacturer’s protocols.

### 2.8. Western Blots

Frozen tissue (100 mg) was homogenized in 1 mL of lysis buffer containing RIPA, protease, and phosphatase inhibitors. The lysate was centrifuged, and the supernatant was collected. Protein concentration was measured using a BCA assay, and samples were diluted in SDS buffer, boiled, and ran on 7% SDS-PAGE gels. After transferring to nitrocellulose membranes, blocking, and overnight incubation with primary antibodies, membranes were treated with secondary antibodies and visualized with ECL substrate.

### 2.9. Realtime-PCR

Total RNA was extracted from liver tissue using TRIzol reagent, and the concentration was measured in triplicate with a NanoDrop ND-1000 spectrophotometer (Sigma-Aldrich, Darmstadt, Germany). One microgram of total mRNA was amplified, and reverse transcribed with the ReadyScript^®^ cDNA Synthesis Mix (Sigma-Aldrich, Darmstadt, Germany), and all reactions were performed in triplicate on a Bio-Rad real-time PCR machine. Statistical analysis was conducted using the ΔΔCT method [34]. The primers used in this study can be found in Table 1.

### 2.10. S-Adenosyl-Methionine (SAM) and DNA Methyltransferase Activity

SAM-dependent methyltransferases were assessed using the Methyltransferase Colorimetric Assay Kit (Cayman Chemical, Ann Arbor, MI, USA), following the manufacturer’s instructions. Total DNMT activity was evaluated with the EpiQuik DNA Methyltransferase Activity/Inhibition Assay Kit (Epigentek, Brooklyn, NY, USA), as previously described [14].

### 2.11. Global DNA Methylation Level Measurement

Genomic DNA was extracted using a Quick-DNATM Universal Kit (Zymo Research, Irvine, CA, USA) as previously described [14].

### 2.12. Whole Genome Bisulfite Sequencing (WGBS)

DNA was extracted from mouse liver samples by using Quick-DNA Miniprep Plus Kit (Zymo Research) and concentrations were tested by Qubit. Bisulfite conversion was conducted according to the protocol of the EZ DNA Methylation-Lightning Kit (Zymo Research) and the target regions were amplified using the Qiagen PyroMark PCR kit. The purity of the PCR product was checked by running a TAE-buffered agarose gel. The products of PCR were purified by Zymoclean Gel DNA Recovery Kits and the concentrations were determined by Qubit. The NEB Next^®^Ultra II DNA Library Prep Kit for Illumina^®^ was used to prepare libraries. In the end, the libraries were pooled together and prepared for sequencing according to the NextSeq System Denature and Dilute Libraries Guide.

### 2.13. Bioinformatic Processing

The WGBS reads were evaluated using software fastQC V0.12.1. All qualified DNA sequenced reads were aligned to the Human Reference Genome Build GRCm38 (mm10) via bwa-meth (version 0.2.2). MethylDackel (version 0.5.2) was used to extract methylation levels from each sample. The mapping quality and sequencing coverage were assessed through samtools flagstat and NGSEP CoverageStats, respectively. The methylation level at each CpG site was calculated by the proportion of methylated reads relative to the total read count. The DNA methylation regions were determined by clustering individual CpG sites using an in-house mean shift-based machine learning program in window size 200. This method iteratively shifts single CpG sites towards the highest density window to identify the local maxima of mean methylation levels.

### 2.14. Statistical Analysis

A one-way analysis of variance (ANOVA) was performed for a significance test between the two groups. A *p* value less than 0.05 was significant. For the data collected from the next-generation sequencing, the methylation plot ranged from −5 kbp to 5 kbp around the transcription start site (TSS) and was smoothed using a 50 bp window for every single site. DMRs are defined as a series of consecutive sites with methylated cytosine. A logistic regression test was used to compare the fractions of methylated cytosines between different groups. A *p*-value < 0.05 was considered a significant difference, while a *p*-value < 0.1 was considered to have marginal significance. All analyses were carried out using R Version R 3.5.2.

## 3. Results

### 3.1. The Treatment with 5-Aza-CdR Successfully Prevented Increased Body Weight, Glucose Intolerance, and MAFLD in Offspring Mice Exposed to Maternal Overnutrition

In the study, all experimental mice exhibited similar body weights at weaning. The group subjected to the HFD continued to gain body weight over the 13-week experimental period. However, upon exposure to 5-Aza-CdR at week 11, the AZA + HFD group displayed a gradual increase in body weight, reaching a plateau one week after AZA + HFD treatment. As a result, starting at week 12, the HFD mice were notably heavier than the AZA + HFD group (Figure 1A). When assessing body weight gain between week 11 and week 13, it was observed that the AZA + HFD group experienced a decline in body weight at week 12 and week 13, whereas the HFD group consistently and progressively gained body weight during this period (Figure 1B). Organ weights were also measured (Figure 1C). Corresponding to their lower body weight, the AZA + HFD group exhibited significantly reduced liver weight, and smaller epididymal fat pads compared to the HFD mice.

We performed an IPGTT to evaluate glucose tolerance in mice at the 10-week mark after weaning (refer to Figure 1D). Following the glucose administration, all mice reached their highest glucose concentrations at the 30 min mark, after which their levels gradually declined over the next 90 min. Notably, the glucose level in the HFD mice did not return to baseline at the 120 min mark, while the AZA + HFD group’s glucose levels did. An analysis of the area under the curve (AUC) revealed that the AZA + HFD group had a significantly lower AUC compared to the HFD group (Figure 1E). Additionally, we measured serum NPRC levels, which showed a slight increasing trend in the AZA + HFD group (Figure 1F). Collectively, these findings suggest that the AZA + HFD mice exhibited improved insulin sensitivity compared to the HFD mice.

Subsequently, we examined liver histology. The livers from the HF diet group exhibited hepatic steatosis, characterized by prominently ballooning cells, macrosteatosis, and hepatocyte degeneration. In contrast, the 5-Aza-CdR-treated group exhibited minimal hepatocyte lipid accumulation, with nuclei generally remaining centrally positioned and occasionally indented due to the presence of small lipid droplets. Some liver cells exhibited slight cytoplasmic swelling but lacked signs of steatosis (Figure 2A). These distinct histological findings align with our measurements of TG content (Figure 2B), indicating that the AZA + HFD group had a lower hepatic TG content, albeit with a significant trend compared to the HFD group (*p* = 0.0781).

Moreover, the serum lipid panel analysis revealed a notably elevated level of HDL in the AZA + HFD group compared to the HFD group (*p* < 0.01, Figure 2D). However, there were no significant differences in the levels of serum total cholesterol, LDL, and VLDL between these two groups (Figure 2C,E).

### 3.2. The AZA Treatment Activated AMPK Signaling and Enhanced PPAR-α Expression in the Offspring’s Liver

Stimulation of AMPK in the liver promotes fatty acid oxidation while inhibiting lipogenesis. Notably, the AZA-treated mice exhibited a significant increase in the p-AMPK-β/AMPK-β ratio, indicating the activation of the AMPK signaling pathway (Figure 3A). Furthermore, PPAR-α, a key positive regulator of fatty acid oxidation, was found to have significantly higher expression levels at both the protein and RNA levels in the AZA + HFD treated group (Figure 3A,B). Additionally, an elevated expression of Insr was also observed in the livers of AZA + HFD-treated mice (Figure 3B), consistent with a slight increasing trend in IRS2 expression (*p* value is 0.1, Figure 3C). These findings collectively suggest an enhancement in fatty acid oxidation and insulin sensitivity in the mice treated with 5-Aza-CdR.

### 3.3. The AZA Treatment Activated Hepatic P-38 and JNK Signaling, Associated with Expression Changes in Genes Involved in Cell Survival and Proliferation

5AZA-CdR is a highly effective antileukemic agent in animal models, primarily through targeting cell cycle progression, specifically the S-phase [32]. Consequently, we investigated the signaling pathways that are critical for cell survival and proliferation. Our analysis revealed an increased level of P-p38/p38 and P-JNK/JNK, indicating the activation of these two signaling pathways in the AZA + HFD group (Figure 4A,B). Furthermore, we examined the hepatic expression of key genes involved in apoptosis (including Bax, Bid, Casp2, Casp3, Casp8, Casp9, Fas, and Ifn-γ) and cell proliferation (including Atm, Cdk2, Cdk4, Cdk6, Chk1, Chk2, P21, CycD1, and CycD2) in the AZA + HFD group compared to the HFD group. The proapoptotic genes, Bax and Casp3, as well as Ifn-γ, exhibited marked increases in the 5-Aza-CdR treated mice (Figure 4C). Additionally, with the 5-Aza-CdR treatment, there was a significant elevation in mRNA levels of proliferative genes, such as Atm, Cdk2, Cdk4, and CycD2 (Figure 4D).

These results collectively suggest a higher rate of cell apoptosis and proliferation in the livers of the AZA + HFD group compared to the HFD group.

### 3.4. The AZA Treatment Enhanced Inflammation in the Offspring’s Liver

In comparison to the HFD group, the mice treated with 5-Aza-CdR exhibited a higher level of hepatic macrophage infiltration, as indicated by an increased presence of F4/80+ cells in the liver (Figure 5A). We also observed a higher ratio of p-NF-κB/NF-κB, signifying the activation of the NF-κB signaling pathway in the AZA + HFD-treated livers (Figure 5B). Consistently, the hepatic expression of Il-1β, Il-6, and Tnfa was significantly elevated in the 5-Aza-CdR-treated mice exposed to an HFD (Figure 5C, *p* < 0.01). However, there were no observable expression differences in Il-8 and Il-10 between these two groups. Collectively, these findings suggest an increased level of hepatic inflammation in the AZA + HFD group.

### 3.5. The AZA Treatment Inhibited DNA Methyltransferase Activation and Thus Decreased the Global DNA Methylation in the Offspring’s Liver

Several studies have indicated that MAFLD induced by an HFD can be ameliorated through folic acid supplementation, possibly by modifying DNA methylation [35,36,37]. Therefore, we examined whether 5-Aza-CdR treatment might influence the expression of genes related to folic acid metabolism. The results showed that none of the genes, including Mtr, Shmt1, Shmt2, Mthfr, Mthfd2, Mthfd2, Tyms, and Dhfr, had changes in their expression under the AZA + HFD treatment compared to the HFD treatment (Figure 6A). This suggests that 5-Aza-CdR itself did not have an impact on folic acid metabolism. As 5-Aza-CdR is a known DNMT inhibitor, it was expected that the DNMT activity would be significantly reduced in the AZA + HFD-treated liver compared to the HFD-treated liver. No significant difference was found in the activity of SAM transferase between the AZA + HFD and HFD groups (Figure 6B). Consistent with the decreased DNMT activity (Figure 6E), the global methylation, indicated by 5 mC%, was significantly lower in the AZA + HFD-treated liver than in the HFD-treated liver (Figure 6D). Importantly, this effect was not attributable to changes in the expression of Dnmt1, Dnmt3a, or Dnmt3b in the AZA + HFD-treated liver (Figure 6C).

### 3.6. Genes Involved in Cell Survival and Inflammation, Whose Hepatic Expression Was Associated with DNA Modification by 5-Aza-CdR, Were Identified in the Liver

We performed whole genome bisulfite sequencing (WGBS) on liver genomic DNA to determine if 5-Aza-CdR treatment altered the methylation profiles of relevant genes. We focused on analyzing the changes in DNA methylation patterns within a 10 kb region, including 5 kb upstream and 5 kb downstream of the transcription start sites (TSS), and compared the results between the AZA + HFD and HFD groups.

Our analysis identified a DMR located 634 bp upstream of the TSS of Bax, exhibiting 53.5% hypomethylation due to 5-Aza-CdR treatment (Figure 6F and Table 2). Additionally, we found a hypermethylated DMR (50% increase) located around 2224 bp upstream of the Casp3 TSS in the AZA + HFD group compared to the HFD group (Figure 6G and Table 2). For Il6, three DMRs were identified due to 5-Aza-CdR treatment, including one hypomethylated region and two hypermethylated regions (Figure 6H and Table 2). The methylation modification levels of these three DMRs were all above 50%.

## 4. Discussion

DNA methylation is a crucial epigenetic modification within mammalian genomes, significantly influencing gene expression by altering the methylation levels at CpG dinucleotides [38]. This process plays a vital role in various biological functions, including cellular differentiation and development, and is implicated in several diseases when dysregulated [38]. Previous research has suggested that a high methylation status increases the risk of obesity and related metabolic disorders, including MAFLD [14,39,40,41]. We have recently reported that the maternal HF diet, initiated before pregnancy and persisting through gestation and lactation, results in a disruption of the methionine cycle, associated with MAFLD in the offspring [14]. This maternal HF diet also results in the global hypermethylation of DNA and targeted DNA methylation modifications on multiple genes involved in one-carbon metabolism and lipid metabolism in the offspring’s liver [14,41]. In this study, we investigated the impact of modifying DNA methylation via 5-Aza-CdR in MAFLD and elucidated its underlying molecular mechanisms. 5-Aza-CdR reduced the genomic DNA methylation level in the liver as expected. We further showed that 5-Aza-CdR counteracted hyperlipidemia and improved glucose tolerance, rescuing subjects from obesity and MAFLD. While 5-Aza-CdR’s role in modifying DNA methylation has been proven with ample evidence in cancer treatment [32,33], it has never been investigated in MAFLD management. Our findings suggested that reversing liver DNA hypermethylation can prevent MAFLD in offspring exposed to maternal overnutrition, underscoring a potential preventive strategy for managing these chronic metabolic conditions in populations with high risks.

Previously, comprehensive genome-wide expression analyses, both in vitro and in vivo, have unveiled that genes associated with apoptosis and cell death are among the most profoundly impacted targets of 5-Aza-CdR [42,43]. In our investigation, we observed an upregulation of apoptotic markers in the livers treated with 5-Aza-CdR and identified DMRs in two apoptotic genes, Bax and Casp3. These findings shed light on the consequences of 5-Aza-CdR-induced DNA methylation alterations in the context of fatty liver. It is plausible that the activation of apoptosis may lead to elevated energy expenditure, potentially promoting fatty acid oxidation, a concept supported by previous studies [44,45]. Notably, while an HFD can disrupt lipid metabolism signaling, resulting in fatty liver due to the inactivation of AMPK and AKT [46], our study revealed the activation of the AMPK signaling pathway and an increase in the expression of PPARα at both the mRNA and protein levels under AZA treatment. These changes align with the observed reduction in hepatic triglyceride levels and the improved state of MAFLD in the AZA + HFD group. Additionally, the overexpression of genes associated with cell proliferation suggests an activated cell proliferation process. Given that 5-Aza-CdR is known to possess potent anti-proliferative effects in both human and animal studies [47,48,49,50,51], it is less likely that the higher expression of proliferation-related genes is a direct effect of 5-Aza-CdR, but rather an apoptosis-induced compensatory proliferation (AiP) [52]. It is important to acknowledge that we were unable to identify DMRs of genes involved in fatty acid oxidation and cell proliferation specifically affected by the low dose of 5-Aza-CdR. This also indirectly supports our assumptions that enhanced proliferation and fatty acid oxidation are induced by increased cell apoptosis. Nonetheless, further research is needed to elucidate the details of this molecular mechanism through 5-Aza-CdR.

It has been well-documented that DNA methylation serves as an epigenetic mechanism that regulates inflammatory responses by influencing the production of pro-inflammatory cytokines [53]. DNA methylation can directly impact the transcription of pro-inflammatory cytokines such as IL-6, IL-8, and TNF-α by modifying the methylation levels of gene promoters [53,54,55,56]. Notably, prior research conducted by the groups of Shi and Xue has indicated that 5-Aza-CdR promotes the polarization of macrophages toward the M2 phenotype by downregulating PPAR-γ due to DNA hypermethylation [57,58]. However, our study revealed a significant increase in the inflammatory state in the liver following treatment with 5-Aza-CdR. This was evidenced by the recruitment of macrophages, the overexpression of pro-inflammatory cytokines, the overactivation of JNK signaling, and the absence of changes in anti-inflammatory biomarkers. As a result, our findings suggest that the macrophages in the fatty liver treated with 5-Aza-CdR tend to exhibit a pro-inflammatory phenotype, which contradicts previous findings [57,58]. It is worth noting that this inconsistency might be attributed to differences in the disease models (obesity and atherosclerosis vs. MAFLD), distinct macrophage types (adipose tissue macrophages vs. hepatic macrophages), and the varying treatment durations of 5-Aza-CdR (3 times per week for 8 weeks vs. daily for 3 weeks) utilized in the studies. Nevertheless, we identified DMRs of Il6 induced by 5-Aza-CdR, establishing a direct link between 5-Aza-CdR treatment and potentially heightened inflammation in the liver. Additionally, we should recognize that an increased apoptotic signal is a potent driver of sterile inflammation in the liver [59], which aligns with our findings of increased cell apoptosis in the HFD + AZA liver. Nonetheless, our discovery regarding 5-Aza-CdR-related hepatic inflammation is cause for concern, despite the beneficial effects in preventing body weight gain and glucose intolerance, improving MAFLD, and lowering blood triglyceride levels. Future animal studies will focus on assessing the long-term effects of 5-Aza-CdR on liver pathology, addressing the crucial issue of its safety in MAFLD treatment.

## 5. Conclusions

The findings of this study highlight a novel epigenetic approach for mitigating the adverse metabolic impacts of maternal overnutrition on offspring. By using 5-Aza-CdR, a DNA methylation inhibitor, the research demonstrates promising results in countering MAFLD in offspring. The observed improvements in weight regulation, glucose homeostasis, and hepatic fat reduction suggest potential for 5-Aza-CdR to influence key metabolic pathways, including AMPK signaling and fatty acid oxidation. However, the increased hepatic inflammation associated with the treatment underscores the complexity of epigenetic modulation and its dual impacts on metabolic health.

This research provides a foundation for translating epigenetic interventions into clinical strategies to address intergenerational health challenges. Future investigations should focus on optimizing treatment protocols, evaluating safety, and assessing long-term outcomes to ensure that epigenetic therapies can be effectively and safely integrated into preventive or therapeutic approaches for MAFLD and related metabolic disorders.

## Figures and Tables

**Figure 1 nutrients-17-00106-f001:**
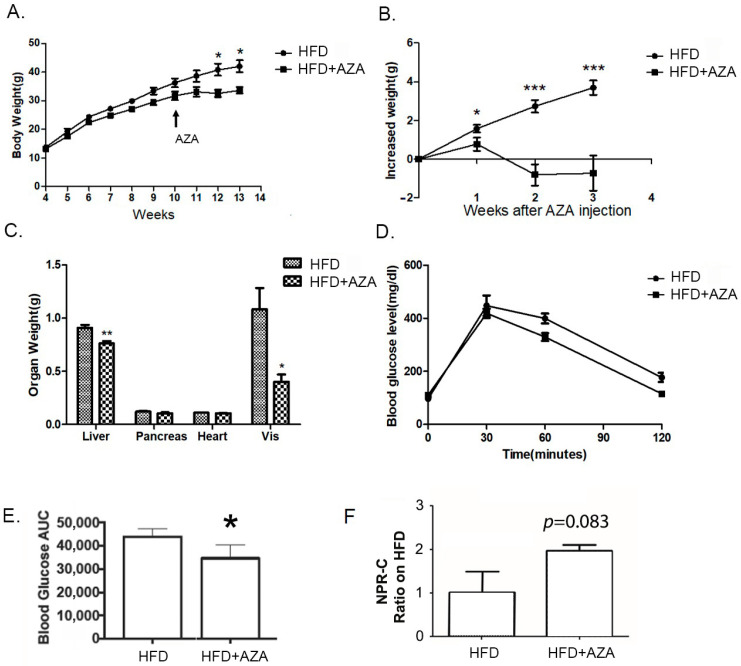
The HF diet resulted in increased body weight and glucose intolerance, while the treatment with 5-Aza-CdR successfully prevented these alterations. (**A**) Body weights of male mice treated with a high-fat diet were recorded weekly after weaning. PBS or 5-Aza-CdR was injected from week 11 to week 13. (**B**) The increased body weights between the high-fat diet group (HFD) and the high-fat diet with the 5-Aza-CdR group (HFD + AZA) were calculated after 5-Aza-CdR was given. (**C**) The weight differences in liver, kidney, and visceral fat between the HF group and the HF + AZA group at the end of week 13 are displayed. (**D**) IPGTT was conducted after week 13. (**E**) The areas under the curve (AUCs) for the IPGTT were calculated. (**F**) Serum NPR-C expression levels were collected and tested in two groups. Data are presented as Mean ± SE, *n* = 10. * Indicates *p* < 0.05 vs. the HF group. ** Indicates *p* < 0.01 vs. the HF group. *** indicates *p* < 0.001 vs. the HF group.

**Figure 2 nutrients-17-00106-f002:**
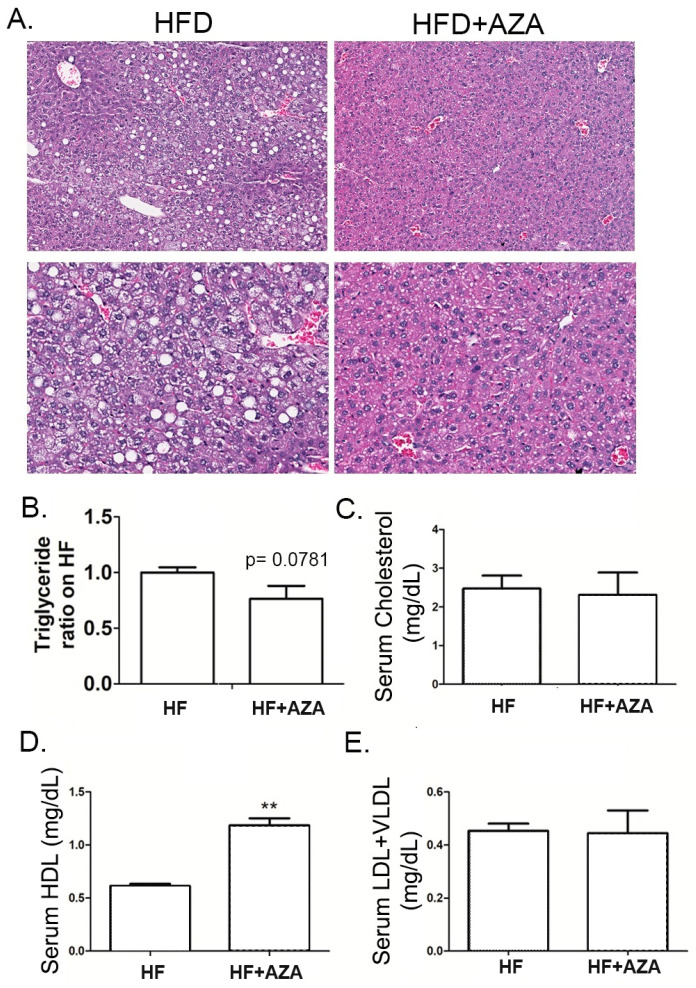
The treatment with 5-Aza-CdR successfully prevented MAFLD in offspring mice exposed to maternal overnutrition. (**A**) H&E staining for the liver tissue in the HF group and HF + AZA group was conducted. (**B**) Hepatic triglyceride concentrations were determined in these two groups. (**C**–**E**) Serum lipid concentrations were detected in these two groups. Data are presented as Mean ± SE, *n* = 5. ** indicates *p* < 0.01 vs. the HF group.

**Figure 3 nutrients-17-00106-f003:**
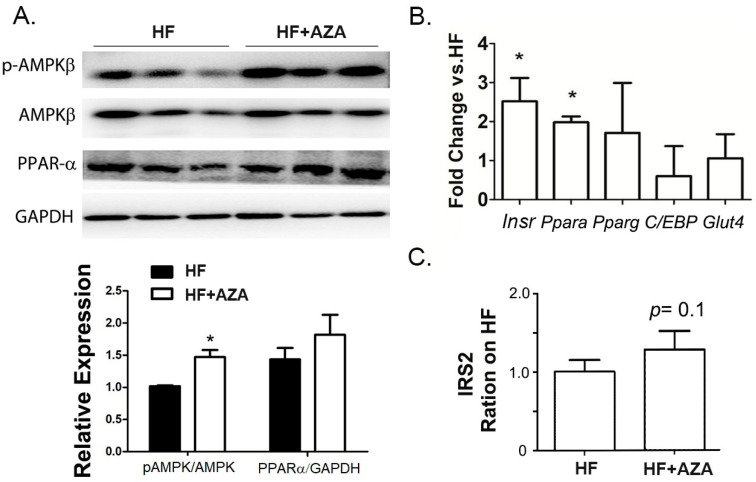
The AZA treatment activated AMPK signaling and enhanced PPAR-α expression in the liver compared to the HFD treatment. (**A**) The expression levels of AMPK-β, phospho-AMPK-β (Ser182), and PPAR-α were detected by Western blot in the livers of mice. Their ratio and relative expression levels were calculated. (**B**) The expression of key genes involved in fatty acid oxidation was measured via real-time PCR. (**C**) Serum IRS2 expression levels were collected and tested in two groups. Data are presented as Mean ± SE, *n* = 5. * indicates *p* < 0.05 vs. the HF group.

**Figure 4 nutrients-17-00106-f004:**
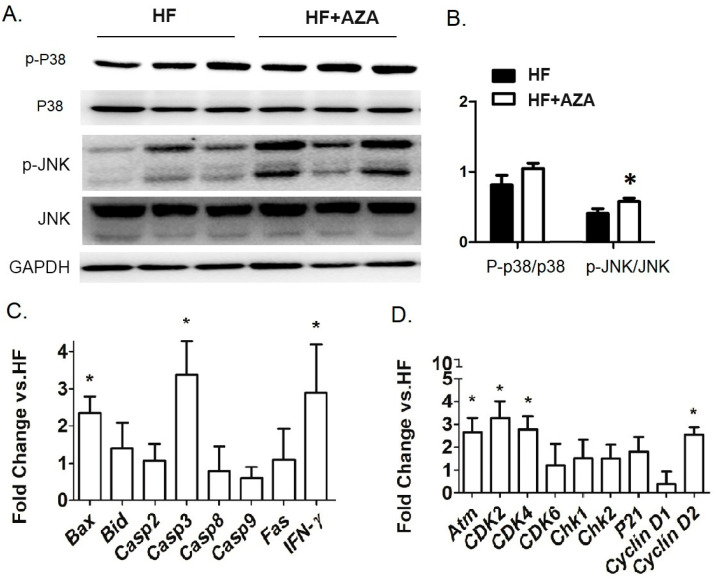
The treatment with AZA resulted in the activation of hepatic P-38 and JNK signaling pathways, which were linked to alterations in the expression of genes related to cell survival and proliferation compared to the HFD group. (**A**) Western blot analysis in liver tissues measured the expression levels of P38 MAPK, phosphorylated P38 MAPK, JNK, and phosphorylated JNK. (**B**) The ratios of phospho-P38 MAPK to total P38 MAPK and phospho-JNK to total JNK were calculated. (**C**,**D**) Key genes involved in cell survival and proliferation were quantified using real-time PCR. Data are presented as Mean ± SE, *n* = 5. * indicates *p* < 0.05 vs. the HF group.

**Figure 5 nutrients-17-00106-f005:**
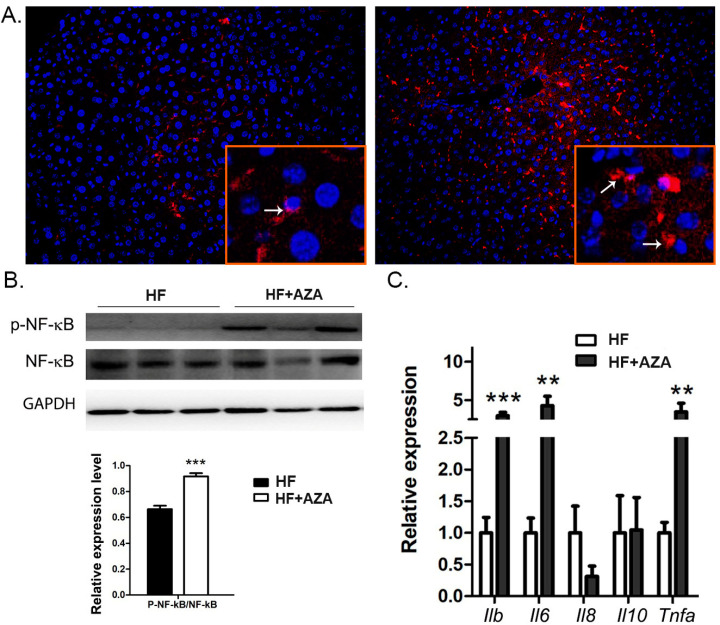
Upon HF and 5-Aza-CdR treatment, inflammation was increased in the liver. (**A**) Immunofluorescence staining of macrophages in mice liver by using F4/80 antibody is shown. Array shows the stained macrophage. (**B**) The expression levels of NF-κB and p-NF-κB were detected by Western blot in mice livers, and their ratio was calculated. (**C**) Key genes involved in inflammation were quantified via real-time PCR. Data are presented as Mean ± SE, *n* = 5. ** indicates *p* < 0.01 vs. the HF group. *** indicates *p* < 0.001 vs. the HF group.

**Figure 6 nutrients-17-00106-f006:**
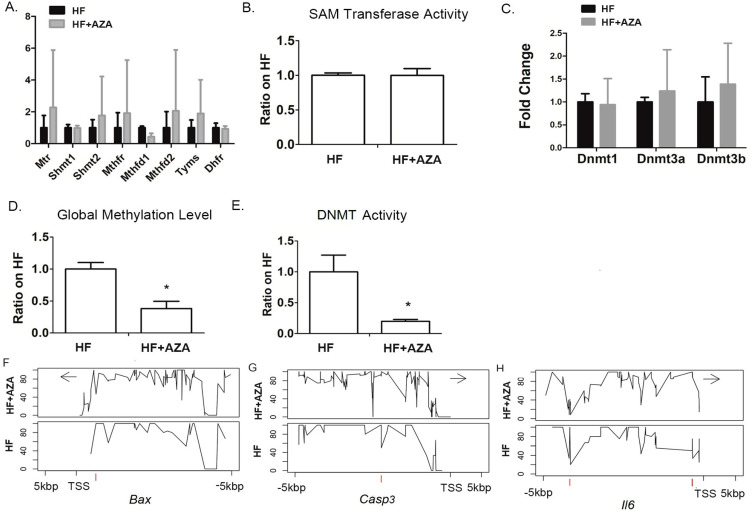
DNA methylation analysis of the livers was conducted between the HFD treatment and the HFD combined with 5-Aza-CdR treatment, and the key genes whose hepatic expression was associated with DNA modification by 5-Aza-CdR were identified in the liver. (**A**) Expression of key genes in folic acid metabolism was measured by real-time PCR in mice livers. (**B**) The enzymatic activity of SAM-dependent methyltransferases was determined in mice livers. (**C**) Hepatic gene expression levels of Dnmt1, Dnmt3a, and Dnmt3b were assessed by real-time PCR. (**D**) Genomic DNA methylation levels were determined in mice livers. (**E**) The enzymatic activity of DNMT was determined in mice livers. (**F**–**H**) The methylation landscape of a 10 kb genomic segment was analyzed through targeted sequencing, encompassing a region from 5 kb upstream to 5 kb downstream of the transcription start site (TSS). The *y*-axis illustrates the methylation percentage at individual nucleotides, while the *x*-axis displays their respective positions relative to the TSS. Data are presented as Mean ± SE, *n* = 5. * indicates *p* < 0.05 vs. the HF group.

**Table 1 nutrients-17-00106-t001:** A list of the primers used in the real-time-PCR.

Primer	Series
Ppara-F	CTGCAGAGCAACCATCCAGAT
Ppara-R	GCCGAAGGTCCACCATTTT
PPARg-F	TGGTTCAAATATGCCACCAG
PPARg-R	CCAAGTGCTGGGATTAAAGG
BAX-F	TGAAGACAGGGGCCTTTTTG
BAX-R	AATTCGCCGGAGACACTCG
BID-F	GCCGAGCACATCACAGACC
BID-R	TGGCAATGTTGTGGATGATTTCT
C/EBPa-F	TGGACAAGAACAGCAACGAG
C/EBPa-R	GCCATGGCCTTGACCAAGGAG
CASP3-F	TGGTGATGAAGGGGTCATTTATG
CASP3-R	TTCGGCTTTCCAGTCAGACTC
CASP8-F	TGCTTGGACTACATCCCACAC
CASP8-R	TGCAGTCTAGGAAGTTGACCA
CASP9-F	TCCTGGTACATCGAGACCTTG
CASP9-R	AAGTCCCTTTCGCAGAAACAG
FAS-F	GGAGGTGGTGATAGCCGGTAT
FAS-R	TGTTTCCACTTCTAAACCATGCT
IFNγ-F	TCAAGTGGCATAGATGTGGAAGAA
IFNγ-R	TGGCTCTGCAGGATTTTCATG
ATM-F	GATCTGCTCATTTGCTGCCG
ATM-R	GTGTGGTGGCTGATACATTTGAT
CDK2-F	CCTGCTTATCAATGCAGAGGG
CDK2-R	TGCGGGTCACCATTTCAGC
CDK4-F	ATGGCTGCCACTCGATATGAA
CDK4-R	TCCTCCATTAGGAACTCTCACAC
CDK6-F	TCCTGCTCCAGTCCAGCTAT
CDK6-R	CCACGTCTGAACTTCCACGA
chk1-F	GTTAAGCCACGAGAATGTAGTGA
chk1-R	GATACTGGATATGGCCTTCCCT
chk2-F	TGACAGTGCTTCCTGTTCACA
chk2-R	GAGCTGGACGAACCCTGATA
P21-F	CCTGGTGATGTCCGACCTG
P21-R	CCATGAGCGCATCGCAATC
Cyclin D2-F	GAGTGGGAACTGGTAGTGTTG
Cyclin D2-R	CGCACAGAGCGATGAAGGT
CyclinD1-F	GCGTACCCTGACACCAATCTC
CyclinD1-R	CTCCTCTTCGCACTTCTGCTC
IL1b-F	GCAACTGTTCCTGAACTCAACT
IL1b-R	ATCTTTTGGGGTCCGTCAACT
IL10-F	GCTCTTACTGACTGGCATGAG
IL10-R	CGCAGCTCTAGGAGCATGTG
IL6-F	TGCCTTCTTGGGACTGATGC
IL6-R	TGGGAGTGGTATCCTCTGTGA
IRS1-F	CTCCTGCTAACATCCACCTTG
IRS1-R	AGCTCGCTAACTGAGATAGTCAT
TNFa-F	AGCCGATGGGTTGTACCTTG
TNFa-R	CCAGACCCTCACACTCAGATC
Shmt1-F	CAGGGCTCTGTCTGATGCAC
Shmt1-R	CGTAACGCGCTCTTGTCAC
Shmt2-F	TGGCAAGAGATACTACGGAGG
Shmt2-R	GCAGGTCCAACCCCATGAT
Mtr-F	GTGCTTGTAAGTCTCCCGTAAG
Mtr-R	GGTTCTGAAAGTCAGGGTCCA
Mthfd1-F	TGAGTGGCTTTGATCCCAATC
Mthfd1-R	GGGAATCCTGAACGGGAAACT
Mthfd2-F	AGTGCGAAATGAAGCCGTTG
Mthfd2-R	GACTGGCGGGATTGTCACC
Tyms-F	GGAAGGGTGTTTTGGAGGAGT
Tyms-R	GCTGTCCAGAAAATCTCG
Mthfr-F	GGCAGCGAGAGTTCCAAGG
Mthfr-R	CAGGGAGAACCACTTGTCACC
Dhfr-F	CGCTCAGGAACGAGTTCAAGT
Dhfr-R	TGCCAATTCCGGTTGTTC
dnmt1-F	CCGTGGCTACGAGGAGAAC
dnmt1-R	TTGGGTTTCCGTTTAGTGGGG
dnmt3a-F	GATGAGCCTGAGTATGAGGATGG
dnmt3a-R	CAAGACACAATTCGGCCTGG
dnmt3b-F	CGTTAATGGGAACTTCAGTGACC
dnmt3b-R	CTGCGTGTAATTCAGAAGGCT
Glut4-F	CCGAAAGAGTCTAAAGCGCCT
Glut4-R	CGTTTCTCATCCTTCAGCTCA

**Table 2 nutrients-17-00106-t002:** Genes with methylation modifications following AZA treatment.

Gene	Dist. to TSS	Chromosome	Position	Meth. Difference *	*p* Value *	AZA	HF
Bax	−634	chr7	45,467,532	53.47222	0.045033	46.52778	100
Casp3	−2224	chr8	46,615,067	−50	0.005057	100	50
Il6	−4173	chr5	30,008,941	58.3333	0.017344	41.66667	100
Il6	−350	chr5	30,012,764	−50	0.017785	100	50
Il6	−346	chr5	30,012,768	−50	0.021717	100	50

* Methylation difference was calculated by subtracting the methylation level of HF from the methylation level of AZA. *p* value < 0.05 is considered a significant difference.

## Data Availability

Data are available upon request.

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
