# Peer review of "Epigenetic Modulation with 5-Aza-CdR Prevents Metabolic-Associated Fatty Liver Disease Promoted by Maternal Overnutrition"

_nutrients, 2024, doi:10.3390/nu17010106_

Round 1
Reviewer 1 Report
Comments and Suggestions for Authors
This study was aimed to investigate whether epigenetic modulation with 5-Aza-CdR, a DNA methylation inhibitor, could prevent metabolically-associated fatty liver disease (MAFLD) in offspring exposed to maternal overnutrition. The results of present study demonstrated that 5-Aza-CdR treatment in HFD-fed offspring effectively countered weight gain, improved glucose regulation, and minimized hepatic fat buildup along with serum lipid imbalances. As the authors concluded, the reviewer thinks that these results suggest a promising epigenetic strategy for MAFLD prevention in offspring affected by in-utero overnutrition. The results of this study suggest that epigenetic modulation with 5-Aza-CdR may be able to prevent MAFLD promoted by maternal overnutrition, and the reviewer considers that this information is extremely useful in view point of preventing MAFLD. Moreover, the reviewer also thinks that the present study is well designed and the analysis is appropriately conducted. However, there are several questions in this study.
The reviewer believes that the progression of MAFLD is due to liver fibrosis. Did the authors evaluate liver fibrosis in this study? If the authors assessed liver fibrosis in this study, please provide the results.
In the “Abstract” section, the authors explain that mice were randomly divided into four groups as follows; PBS, AZA, HFD, or AZA+HFD. However, based on “Methods” and “Results” sections, it appears there were only two groups: HFD and AZA+HFD. Which one is correct?
In relation to the above, in this study, the authors used one-way analysis of variance (ANOVA) to compare the two groups. In generally, one-way ANOVA is used to compare multiple group or time series comparisons, while unpaired t-tests or Mann-Whitney U-tests are used to compare two groups. Furthermore, when using one-way ANOVA, post-hoc tests should be used. The authors should reconsider their statistical analysis.
This article does not mention the number of mice. The authors should explain how many mice they used in this study.
Could the results of this study be applied to humans in the future? Please explain authors’ future outlook for how the results of this study will be applied clinically.
Author Response
Comments 1: The reviewer believes that the progression of MAFLD is due to liver fibrosis. Did the authors evaluate liver fibrosis in this study? If the authors assessed liver fibrosis in this study, please provide the results.
Response1: Thanks for the valuable comments. In general, liver fibrosis develops 16 weeks after HFD exposure (PMID: 35212982), which was supported by our previous reports using the same mouse models as we did in this study (PMID: 36270572 and PMID: 30316166). Our experimental mice were exposed to 60% HFD for only 8 weeks, we did not think that these mice developed fibrosis, therefore, liver fibrosis was not examined.
Comments 2: In the “Abstract” section, the authors explain that mice were randomly divided into four groups as follows; PBS, AZA, HFD, or AZA+HFD. However, based on “Methods” and “Results” sections, it appears there were only two groups: HFD and AZA+HFD. Which one is correct?
Response2: Sorry for the confusion, this mistake is corrected in the revised abstract.
Comments 3: In relation to the above, in this study, the authors used one-way analysis of variance (ANOVA) to compare the two groups. In generally, one-way ANOVA is used to compare multiple group or time series comparisons, while unpaired t-tests or Mann-Whitney U-tests are used to compare two groups. Furthermore, when using one-way ANOVA, post-hoc tests should be used. The authors should reconsider their statistical analysis.
Response3: Thank you for your comment regarding our analytical approach. We appreciate the opportunity to clarify our choice of using one-way ANOVA which was performed by our biostatistician Kurt Zhang (https://ibt.tamu.edu/faculty/kurt-zhang.html) . It is important to note that when comparing means between two groups, one-way ANOVA and unpaired t-tests are statistically equivalent. This equivalence arises because both tests evaluate the same null hypothesis—that the group means are equal—and they both rely on the assumption of normality to compute the test statistic. In the case of two groups, the F-statistic from the ANOVA becomes squared of the T-statistic from the t-test, leading to the same p-value. Therefore, while one-way ANOVA is typically used for comparisons across more than two groups, its application to just two groups provides the same inferential strength and validity as the unpaired t-test. We opted for ANOVA due to its flexibility and familiarity with our analysis framework.
Comments 4: This article does not mention the number of mice. The authors should explain how many mice they used in this study.
Response4: This information is now added to the revised section “ 2.1. Animal study”.
Comments 5: Could the results of this study be applied to humans in the future? Please explain authors’ future outlook for how the results of this study will be applied clinically.
Response5: We thank the insightful comments that significantly improved the quality of our manuscript. We now have added a translational perspective of the study in the section “5. Conclusions”.
Reviewer 2 Report
Comments and Suggestions for Authors
The manuscript has evaluated the role of 5-Aza-CdR in preventing metabolic-associated fatty liver disease promoted by maternal overnutrition involving inhibition of hypermethylation. The manuscript is well written, and the conclusion is supported by the data included in the manuscript. I have following concerns.
1. The results (line 64 -72) should not be included in the introduction. These should be in the results or the conclusion section.
2. Please format the manuscript. The subsection and the figure legends are not as per the manuscript style.
3. Animal study section, the housing conditions, number of pregnant females, number of males, and the litter size must be mentioned.
4. Please include the forward and reverse sequence of primers used for RT-PCR.
5. Methylation and demethylation with and without 5-Aza-CdR should be supported by in-vitro experiments.
6. In the end, the libraries were pooled together and prepared for sequencing according to the NextSeq System Denature and Dilute Libraries Guide- details of sequencing pipeline and methodology should be included in the manuscript.
7. Figure 5A- immunofluorescence staining is not evident, please provide better images and point out the staining (red with DAPI) using an arrow.
Author Response
Comments 1: The results (line 64 -72) should not be included in the introduction. These should be in the results or the conclusion section.
Response 1: Per suggestion, this part is removed from the revised manuscript.
Comments 2: Please format the manuscript. The subsection and the figure legends are not as per the manuscript style.
Response 2:To address this comment, we are seeking further guidance from the journal editors and have already reached out to them. Once we receive their feedback, we will make the necessary revisions to align with the journal's requirements.
Comments 3: Animal study section, the housing conditions, number of pregnant females, number of males, and the litter size must be mentioned.
Response 3: We agree with the reviewer’s valuable comments. This information are now added to the revised section “ 2.1. Animal study”.
Comments 4: Please include the forward and reverse sequence of primers used for RT-PCR.
Response 4: Per suggestion, a list of the primers used for realtime PCR is now available in Table 1.
Comments 5: Methylation and demethylation with and without 5-Aza-CdR should be supported by in-vitro experiments.
Response 5: We appreciate the feedback and acknowledge that a limitation of the current study is the lack of direct evidence linking DNA methylation at the identified sites to the expression changes of apoptotic and inflammatory genes. Due to the sequencing depth, we were only able to detect a limited number of differentially methylated regions (DMRs) associated with these genes. It is well-established that modifications across multiple DMR sites work synergistically to influence gene expression. Therefore, while the current study suggests an association between these DMR sites and gene expression changes, it does not establish causality. Future studies will address this by increasing sequencing depth to identify additional DMRs and conducting in vitro validation assays to provide more direct evidence of these relationships.
Comments 6: In the end, the libraries were pooled together and prepared for sequencing according to the NextSeq System Denature and Dilute Libraries Guide- details of sequencing pipeline and methodology should be included in the manuscript.
Response 6:Thank you for your comment requesting more detail on our bioinformatics procedure. In response, we have added a new section (2.13) of Bioinformatics processing which gives a comprehensive description of our pipeline and analysis methodology to the WGBS data. We believe that these additions will provide clarity and enhance the reproducibility of our study. We appreciate your comment. The updated information can be found in the Methods section of the manuscript.
Comments 7: Figure 5A- immunofluorescence staining is not evident, please provide better images and point out the staining (red with DAPI) using an arrow.
Response 7: Better images pointing out the staining are now added to the revised manuscript.
Reviewer 3 Report
Comments and Suggestions for Authors
The study was well prepared in revealing the interesting results of epigenetic changes in fatty liver, while the attitude toward a marginal significance was recommended to be modified. It is also better to re-review carefully the abbreviations and technical terms of the overall text to improve the quality of the paper.
1. What is the therapeutic way of reversing liver DNA methylation? The authors’ ideas are expected to be more described based on some evidence-based references in Discussion.
2. Typically, a marginal significant level can not be defined in an academic policy. That is a subjective manner only. The defined part of marginal significant would be removed. The authors may use the term ‘marginally’ or ‘significant trend’ without such a subjective definition’ in Results and Discussion.
3. MAFLD was used in Title and Introduction, while NAFLD was also seen in the text. The technical terms could be unified.
4. Abstract; Conclusions did not appear to be concrete. It should describe the more ‘concrete’ suggestion based the current study’s results.
5. Line 20; the term ‘CD’ is not required to be abbreviated because CD was not used in Abstract.
6. Line 30; the term ‘DMRs’ is not required to be abbreviated because DMRs was not used in Abstract.
7. Line 48; the term ‘themselves’ may not be proper; then, itself can be used instead.
8. There was found to be the term ‘HF diets’ in Line 55, ‘HFD’ (which was not abbreviated in the text) in line 56, and ‘HF diet’ in line 66. The unified expression may be proper (it may be hard to read).
9. Line 61; what is DNMT? Fully spelling out might be better in this part, instead of the abbreviation in the latter part.
10. Line 72; the subheading ‘2….’ Was misplaced.
11. Line 80; the ethical approval No. of University could be added.
12. Line 105; LDL/VLDL could be fully spelled out before the abbreviation.
13. Line 105; the assay performance (e.g., accuracy, coefficient of variation) could be described.
14. Line 170; the use of AUC analysis could be described in the part of ‘2.13. Statistics’.
15. Line 165; IPGGT was already abbreviated.
16. Line 195; TG can be abbreviated at the first appearance.
17. Line 306; DMR was already abbreviated.
18. Line 320; the end of the sentence can have some references.
19. Line 322; the end of the sentence can have some references.
20. Line 368; the end of the sentence can have some references.
21. The term ‘in vivo’ and ‘in vitro’ could be expressed in an Italic style.
Author Response
Comments 1: What is the therapeutic way of reversing liver DNA methylation? The authors’ ideas are expected to be more described based on some evidence-based references in Discussion.
Response 1: We thank the insightful comments that significantly improved the quality of our manuscript. 5-Aza-CdR incorporates into DNA, traps DNMTs, and leads to passive demethylation during cell replication, which has been proved with ample evidence in cancer treatment (PMID: 16015507 and PMID: 21965114) but has never been investigated in MAFLD. Therefore, we carried out this novel study to explore whether a low dose of 5-Aza-CdR could mitigate MAFLD with a high risk due to maternal overnutrition. We have included the reference in our study and also have added a translational perspective of the study in the section “5. Conclusions”.
Comments 2: Typically, a marginal significant level can not be defined in an academic policy. That is a subjective manner only. The defined part of marginal significant would be removed. The authors may use the term ‘marginally’ or ‘significant trend’ without such a subjective definition’ in Results and Discussion.
Response 2: We fully concur with the reviewer’s observation regarding the use of 'marginal significance.' We did not intend to present marginal significance as a definitive finding. Instead, it was mentioned merely to indicate a possible trend that could warrant further investigation with a larger sample size. We appreciate the guidance to use terms like 'marginally' or 'significant trend' and have changed the corresponding texts in the Results and Discussion sections.
Comments 3: MAFLD was used in Title and Introduction, while NAFLD was also seen in the text. The technical terms could be unified.
Response 3: We have unified to “MAFLD” in the revised manuscript.
Comments 4: Abstract; Conclusions did not appear to be concrete. It should describe the more ‘concrete’ suggestion based the current study’s results.
Response 4: The conclusion is changed to “5-Aza-CdR was shown to mitigate MAFLD features in offspring of maternal overnutrition by reversing DNA hypermethylation and improving metabolic pathways, though its dual impact on inflammation highlights the need for further research to optimize its therapeutic potential.”
Comments 5: Line 20; the term ‘CD’ is not required to be abbreviated because CD was not used in Abstract.
Response 5: “CD” is changed to “chow diet” in the revised manuscript.
Comments 6: Line 30; the term ‘DMRs’ is not required to be abbreviated because DMRs was not used in Abstract.
Response 6: “DMRs” is spread out in the revised abstract.
Comments 7: Line 48; the term ‘themselves’ may not be proper; then, itself can be used instead.
Response 7: Corrected as suggested.
Comments 8: There was found to be the term ‘HF diets’ in Line 55, ‘HFD’ (which was not abbreviated in the text) in line 56, and ‘HF diet’ in line 66. The unified expression may be proper (it may be hard to read).
Response 8: Unified as suggested.
Comments 9: Line 61; what is DNMT? Fully spelling out might be better in this part, instead of the abbreviation in the latter part.
Response 9: Revised as suggested.
Comments 10: Line 72; the subheading ‘2….’ Was misplaced.
Response 10: Correction is made as suggested.
Comments 11: Line 80; the ethical approval No. of University could be added.
Response 11: The AUP No. IACUC 2016-0326 is added to the revised manuscript.
Comments 12: Line 105; LDL/VLDL could be fully spelled out before the abbreviation.
Response 12: Added as suggested.
Comments 13: Line 105; the assay performance (e.g., accuracy, coefficient of variation) could be described.
Response 13: As suggested, we added that “The assay demonstrated high accuracy with a coefficient of variation (CV) of less than 5%, ensuring reliable and reproducible results for lipid profiling, according to the manufacturer’s instructions.” To the revised manuscript.
Comments 14: Line 170; the use of AUC analysis could be described in the part of ‘2.13. Statistics’.
Response 14: While the Area under the curve (AUC) is compared between the two groups, we did not use distinct statistical analysis for it.
Comments 15: Line 165; IPGGT was already abbreviated.
Response 15: Revised as suggested.
Comments 16: Line 195; TG can be abbreviated at the first appearance.
Response 16: Revised as suggested.
Comments 17: Line 306; DMR was already abbreviated.
Response 17: Revised as suggested.
Comments 18: Line 320; the end of the sentence can have some references.
Response 18: Added PMID: 24789823
Comments 19: Line 322; the end of the sentence can have some references.
Response 19: Added PMID: 24789823
Comments 20: Line 368; the end of the sentence can have some references.
Response 20: Reference was not added, because this sentence summarizes our findings if there was no misalign for line 368.
Comments 21: The term ‘in vivo’ and ‘in vitro’ could be expressed in an Italic style.
Response 21: Revised as suggested.
Round 2
Reviewer 1 Report
Comments and Suggestions for Authors
I think all responses to reviewers' comments have been addressed satisfactorily.
I have no comments on the revised manuscript.
Author Response
Thank you for your approval!
Reviewer 2 Report
Comments and Suggestions for Authors
Please include the limitations of the study separately in the manuscript as are in the authors reply.
In Figure 5A, colocalized cells have not been images in inset, there are dual positive cells but in B panel, only one cell is in inset while in panel A, no dual positive cells are there in the inset, please focus on the are of dual positive cells and it will be better to count dual positive cells in at least 5 areas of 3 images and plot the average to support the results.
Author Response
Comment 1: In Figure 5A, colocalized cells have not been images in inset, there are dual positive cells but in B panel, only one cell is in inset while in panel A, no dual positive cells are there in the inset, please focus on the are of dual positive cells and it will be better to count dual positive cells in at least 5 areas of 3 images and plot the average to support the results.
Response 1: Thank you for your suggestion. We have updated the images as you recommended and counted the positive cells accordingly. Following your guidance, we quantified the F4/80-positive cells per field and found that the HFD + AZA group had approximately twice the percentage of positive cells compared to the HFD group. We did not include this data in the manuscript as it is relatively subjective. However, our Real-Time PCR results for inflammatory genes are consistent with the IF staining data, and together, they support the conclusion of increased inflammation in the HFD + AZA group.
Reviewer 3 Report
Comments and Suggestions for Authors
The paper was much improved.
Author Response
Thank you for your approval!